# Bayesian Inference of Temporal Specifications to Explain How Plans Differ

**Anonymous for Blind Review**

## Abstract

Temporal logics are useful for describing dynamic system behavior, and have been successfully used as a language for goal definitions during task planning. Prior works on inferring temporal logic specifications have focused on "summarizing" the input dataset – i.e., finding specifications that are satisfied by all plan traces belonging to the given set. In this paper, we examine the problem of inferring specifications that describe temporal differences between two sets of plan traces. We formalize the concept of providing such *contrastive* explanations, then present a Bayesian probabilistic model for inferring contrastive explanations as linear temporal logic specifications. We demonstrate the efficacy, scalability, and robustness of our model for inferring correct specifications across various benchmark planning domains and for a simulated air combat mission.

## 1 Introduction

In a meeting where multiple plan options are under deliberation by a team, it would be helpful for that team's resolution process if someone could intuitively explain how the plans under consideration differ from one another. Also, given a need to identify differences in execution behavior between distinct groups of users (e.g., a group of users who successfully completed a task using a particular system versus those who did not), explanations that identify distinguishing patterns between group behaviors can yield valuable analytics and insights toward iterative system refinement.

In this paper, we seek to generate explanations for how two sets of divergent plans differ. We focus on generating such *contrastive* explanations by discovering specifications satisfied by one set of plans, but not the other. Prior works on plan explanations include those related to plan recognition for inferring latent goals through observations (Ramırez and Geffner 2010; Zhuo 2017), works on system diagnosis and excuse generation in order to explain plan failures (Sohrabi, Baier, and McIlraith 2010; Göbelbecker et al. 2010), and those focused on synthesizing "explicable" plans – i.e., plans that are self-explanatory with respect to a human's mental model (Zhang et al. 2017; Kulkarni et al. 2019). The aforementioned works, however, only involve the explanation or generation of a single plan; we instead focus on explaining differences between multiple plans, which can be helpful in various applications, such as the analysis of competing systems and compliance models, and detecting anomalous behaviour of users.

A specification language should be used in order to achieve clear and effective plan explanations. Prior works have considered surface-level metrics such as plan cost and action (or causal link) similarity measures to describe plan differences (Nguyen et al. 2012; Borgo, Cashmore, and Magazzeni 2018). In this work, we leverage linear temporal logic (LTL) (Pnueli 1977) which is an expressive language for capturing temporal relations of state variables. We use a plan's individual satisfaction (or dissatisfaction) of LTL specifications to describe their differences.

LTL specifications have been widely used in both industrial systems and planning algorithms to compactly describe temporal properties (Yang et al. 2006). They are human interpretable when expressed as compositions of predefined templates; inversely, they can be constructed from natural language descriptions (Dzifcak et al. 2009) and serve as natural patterns when encoding high-level human strategies for planning constraints (Kim, Banks, and Shah 2017).

Although a suite of LTL miners have been developed for software engineering and verification purposes (Yang et al. 2006; Lemieux, Park, and Beschastnikh 2015; Shah et al. 2018), they primarily focus on mining properties that summarize the overall behavior on a single set of plan traces. Recently, Neider and Gavran (2018) presented SAT-based algorithms to construct a LTL specification that asserts contrast between two sets of traces. The algorithms, however, are designed to output only a single explanation, and are susceptible to failure when the input contains imperfect traces. Similar to Neider and Gavran, our problem focuses on mining contrastive explanations between two sets of traces, but we adopt a probabilistic approach — we present a Bayesian inference model that can generate multiple explanations while demonstrating robustness to noisy input. The model also permits scalability when searching in large hypothesis spaces and allows for flexibility in incorporating various forms of prior knowledge and system designer preferences. We demonstrate the efficacy of our model for extracting correct explanations on plan traces across various benchmark planning domains and for a simulated air combat mission.

## 2 Related Work

Plan explanations are becoming increasingly important as automated planners and humans collaborate. This first involves humans making sense of the planner's output (e.g., PDDL plans), where prior work has focused on developing user-friendly interfaces that provide graphical visualizations to describe the causal links and temporal relations of plan steps (Bidot et al. 2010; Seegebarth et al. 2012; Magnaguagno et al. 2017). The outputs of these systems, however, require an expert for interpretation and do not provide a direct explanation as to *why* the planner made certain decisions to realize the outputted plan.

Automatic generation of explanations has been studied in *goal recognition* settings, where the objective is to infer the latent goal state that best explains the incomplete sequence of observations (Ramırez and Geffner 2010; Sohrabi, Riabov, and Udrea 2016). Works on *explicable planning* emphasize the generation of plans that are deemed self-explanatory, defined in terms of optimizing plan costs for a human's mental model of the world (Zhang et al. 2017; Kulkarni et al. 2019). Mixed-initiative planners iteratively revise their plan generation based on user input (e.g. action modifications), indirectly promoting an understanding of differences across newly generated plans through continual user interaction (Sengupta et al. 2017; Borgo, Cashmore, and Magazzeni 2018). All aforementioned works deal with explainability with respect to a single planning problem specification, whereas our model deals with explaining differences in specifications governing two distinct sets of plans given as input.

Works on *model reconciliation* focus on producing explanations for planning models (i.e. predicates, preconditions and effects), instead of the realized plans (Chakraborti et al. 2017). Explanations are specified in the form of model updates, iteratively bringing an incomplete model to a more complete world model. The term, "contrastive explanation," is used in these works to identify the relevant differences between the input pair of models. Our work is similar in spirit but focuses on producing a specification of differences in the constraints satisfied among realized plans. Our approach takes sets of observed plans as input rather than planning models.

While model updates are an important modality for providing plan explanations, there are certain limitations. We note that an optimal plan generated with respect to a complete environment/world model is not always explicable or self-explanatory. The space of optimal plans may be large, and the underlying preference or constraint that drives the generation of a particular plan may be difficult to pre-specify and incorporate within the planning model representation. We focus on explanations stemming directly from the realized plans themselves. Environment/world models (e.g. PDDL domain files) can be helpful in providing additional context, but are not necessary for our approach.

Our work leverages LTL as an explanation language. Temporal patterns can offer greater expressivity and explanatory power in describing *why* a set of plans occurred and *how* they differ, and may reveal hidden plan dynamics that cannot be captured by the use of surface-level metrics like plan cost or action similarities. Our work on using LTL for contrastive explanations directly contributes to exploring how we can answer the following roadmap questions for XAIP (Fox, Long, and Magazzeni 2017): "why did you do that? why didn't you do something else (that I would have done)?"

Prior research into mining LTL specifications has focused on generating a "summary" explanation of the observed traces. Kasenberg and Scheutz (2017) explored mining globally persistent specifications from demonstrated action traces for a finite state Markov decision process. Lemieux, Park, and Beschastnikh (2015) introduced Texada, a system for mining all possible instances of a given LTL template from an output log where each unique string is represented as a new proposition. Shah et al. (2018) proposed a template-based probabilistic model to infer task specifications given a set of demonstrations. However, all of these approaches focus on inferring a specification that all the demonstrated traces satisfy.

For contrastive explanations, Neider and Gavran (2018) presented SAT-based algorithms to infer a LTL specification that delineates between the positive and negative sets of traces. Unlike existing LTL miners, the algorithms construct an arbitrary, minimal LTL specification without requiring predefined templates. However, they are designed to output only a single specification, and can fail when the sets contain imperfect traces (i.e., if there exists no specification consistent with every single input trace.). We present a probabilistic model for the same problem and generate multiple contrastive explanations while offering robustness to noisy input.

Some works have proposed algorithms to infer contrastive explanations for continuous valued time-series data based on restricted signal temporal logic (STL) grammar (Yoo and Belta 2017; Kong, Jones, and Belta 2017). However, the continuous space semantics of STL and a restricted subset of temporal operators make the grammar unsuitable for use with planning domain problems. To the best of our knowledge, our proposed model is the first probabilistic model to infer contrastive LTL specifications for sets of traces in domains defined by PDDL.

## 3 Preliminaries

### Linear Temporal Logic

Linear Temporal Logic (LTL) provides an expressive grammar for describing temporal behavior (Pnueli 1977). An LTL specification $\varphi$ is constructed from a set of propositions $V$, the standard Boolean operators, and a set of temporal operators. Its truth value is determined with respect to a trace, $\pi$, which is an infinite or finite sequence of truth assignments for all propositions in $V$. The notation $\pi, t \models \varphi$ indicates that $\varphi$ holds at time $t$. The trace $\pi$ satisfies $\varphi$ (denoted by $\pi \models \varphi$) iff $\pi, 0 \models \varphi$. The minimal syntax for LTL can be described as follows:

$$\varphi ::= p \mid \neg\varphi_1 \mid \varphi_1 \vee \varphi_2 \mid \mathbf{X}\varphi_1 \mid \varphi_1 \mathbf{U} \varphi_2, \qquad (1)$$

where $p$ is a proposition, and $\varphi_1$ and $\varphi_2$ are valid LTL specifications.

| Template | $n_T$ | Formula | Meaning |
|---|---|---|---|
| $\varphi_{global}$ | 1 | $\mathbf{G}p_i$ | $p_i$ is true throughout the entire trace. |
| $\varphi_{eventuality}$ | 1 | $\mathbf{F}p_i$ | $p_i$ eventually occurs (may later become false). |
| $\varphi_{stability}$ | 1 | $\mathbf{FG}p_i$ | $p_i$ eventually occurs and stays true forever. |
| $\varphi_{response}$ | 2 | $\mathbf{G}(p_i \rightarrow \mathbf{XF}p_j)$ | If $p_i$ occurs, $p_j$ eventually follows. |
| $\varphi_{until}$ | 2 | $p_i\mathbf{U}p_j$ | $p_i$ has to be true until $p_j$ eventually becomes true. |
| $\varphi_{atmostonce}$ | 1 | $\mathbf{G}(p_i \rightarrow (p_i\mathbf{U}(\mathbf{G}\neg p_i)))$ | Only one contiguous interval exists where $p_i$ is true. |
| $\varphi_{sometime-before}$ | 2 | $(p_j \wedge \neg p_i)\mathbf{R}(\neg p_i)$ | If $p_i$ occurs, $p_j$ occurred in the past. |

Table 1: An example set of LTL templates. $n_T$ corresponds to the number of free propositions for each template.

**X** reads as "next" where $\mathbf{X}\varphi$ evaluates as true at $t$ if $\varphi$ holds in the next time step $t + 1$. **U** reads as "until" where $\varphi_1\mathbf{U}\varphi_2$ evaluates as true at time step $t$ if $\varphi_1$ is true at that time and going forward, until a time step is reached where $\varphi_2$ becomes true. In addition to the minimal syntax, we also use higher-order temporal operators, **F** (eventually), **G** (global), and **R** (release). $\mathbf{F}\varphi$ holds true at $t$ if $\varphi$ holds for some time step $\geq t$. $\mathbf{G}\varphi$ holds true at $t$ if $\varphi$ holds for all time steps $\geq t$. $\varphi_1\mathbf{R}\varphi_2$ holds true at time step $t$ if either there exists a time step $t_1 \geq t$ such that $\varphi_2$ holds true until $t_1$ where both $\varphi_1$ and $\varphi_2$ hold true simultaneously, or no such $t_1$ exists and $\varphi_2$ holds true for all time steps $\geq t$.

Interpretable sets of LTL templates have been defined and successfully integrated for a variety of software verification systems (Yang et al. 2006; Maggi, Mooij, and van der Aalst 2011). Some of the widely used templates are shown in Table 1.

### Contrastive Explanation

According to Elzein (2018), a contrastive explanation describes "why event A occurred as opposed to some alternative event B." In our problem, events A and B represent two sets of plan traces (can be seen as traces generated from different systems or different group behavior). The form of *why* may be expressed in various ways (Lombrozo 2006); our choice is to define it according to the plans' satisfaction of a constraint. Then, formally:

**Definition 3.1.** *A contrastive explanation is a constraint $\varphi$ that it is satisfied by one set of plan traces (positive set, $\boldsymbol{\pi_A}$), but not by the other (negative set, $\boldsymbol{\pi_B}$).*

The constraint $\varphi$ can be seen as a classifier trying to separate the provided positive and negative traces. Its performance measure corresponds to standard classification accuracy, computed by counting the number of traces in $\boldsymbol{\pi_A}$ that satisfy $\varphi$ and, conversely, the number of traces in $\boldsymbol{\pi_B}$ where $\varphi$ is unsatisfied. Formally, accuracy of $\varphi$ is:

$$\frac{|\{\pi : \pi \models \varphi, \pi \in \boldsymbol{\pi_A}\}| + |\{\pi : \pi \nvDash \varphi, \pi \in \boldsymbol{\pi_B}\}|}{|\boldsymbol{\pi_A}| + |\boldsymbol{\pi_B}|} \quad (2)$$

Accuracy is 1 for a perfect contrastive explanation, and approaches zero if both sets contains no valid trace with respect to $\varphi$ (i.e., all traces in $\boldsymbol{\pi_A}$ dissatisfy $\varphi$ and all traces in $\boldsymbol{\pi_B}$ satisfy $\varphi$).

## 4 Problem Statement and Approach

The input to the problem is a pair of sets of traces $(\boldsymbol{\pi_A}, \boldsymbol{\pi_B})$. Each $\pi_i \in \boldsymbol{\pi}$ is a trace on the set of propositions $\boldsymbol{V}$ (we refer to $\boldsymbol{V}$ as the vocabulary). The output is a set of specifications, $\{\varphi\}$, where each $\varphi$ achieves perfect or near-perfect contrastive explanation. This is an unsupervised classification problem.

We use LTL specifications for the choice of $\varphi$. Planning is sequential, and so temporal patterns can offer greater expressivity and explanatory power for identifying plan differences rather than static facts. We utilize a set of interpretable LTL templates, such as those shown in Table 1.

A LTL template $T$ is instantiated with a selection of $n_T$ propositions denoted by $\boldsymbol{p} \in \boldsymbol{V}^{n_T}$. The candidate formula $\varphi$ is then composed as a conjunction of multiple instantiations of a template $T$ based on a set of selections $\{\boldsymbol{p}\} \subseteq \boldsymbol{V}^{n_T}$. For example, an instantiation of $T =$"stability" with $\boldsymbol{p} = [apple]$ is written as $\mathbf{FG}(apple)$. If the selected subset of propositions is $\{\boldsymbol{p}\} = \{[apple], [banana], [carrot]\}$, then $\varphi = \mathbf{FG}(apple) \wedge \mathbf{FG}(banana) \wedge \mathbf{FG}(carrot)$, asserting the stability condition for all three propositions. Conjunctions provide powerful semantics with the ability to capture a notion of quantification. Formally, our LTL specification is written as follows:

$$\varphi_T = \bigwedge_{\boldsymbol{p}\in\{\boldsymbol{p}\}} T(\boldsymbol{p}), \quad (3)$$

Note that the number of free propositions, $n_T$, varies per LTL template. The number of possible specifications for a given LTL template $T$ is $2^{|\boldsymbol{V}|^{n_T}}$. Instead of extracting specifications narrowed down to a single template query, our hypothesis space $\boldsymbol{\Phi}$ is set to include a number of predefined templates, $T_1, T_2, ...T_k$. With $k$ representing the number of possible templates, the full hypothesis space of $\boldsymbol{\Phi}$ grows with $\mathcal{O}(k \cdot 2^{|\boldsymbol{V}|^{n_T}})$. Employing brute force enumeration to find $\{\varphi\}$ that achieves the contrastive explanation criterion rapidly becomes intractable with increasing vocabulary size.

### Bayesian Inference

We model specification learning as a Bayesian inference problem, building on the fundamental Bayes theorem:

$$P(\varphi \mid \mathbf{X}) = \frac{P(\varphi)P(\mathbf{X} \mid \varphi)}{\sum_{\varphi\in\boldsymbol{\Phi}} P(\varphi)P(\mathbf{X} \mid \varphi)} \quad (4)$$

Our goal is to infer $\varphi^* = \operatorname{argmax}_{\boldsymbol{\Phi}} P(\varphi|\mathbf{X})$. $P(\varphi)$ represents the prior distribution over the hypothesis space, and

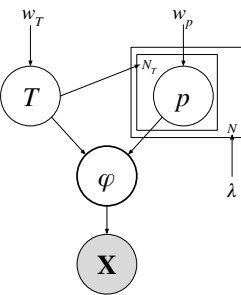

Figure 1: A graphical model of the generative distribution. $\varphi$ represents the latent LTL specification that we seek to infer given the evidence $\mathbf{X}$ (in our case, the traces).

$P(\mathbf{X} \mid \varphi)$ is the likelihood of observing the evidence $\mathbf{X} = (\boldsymbol{\pi_A}, \boldsymbol{\pi_B})$ given $\varphi$. We adopt a probabilistic generative modeling approach that has been used extensively in topic modeling (Blei, Ng, and Jordan 2003). Below, we describe each component of our generative model, depicted in Figure 1.

**Prior Function**  $\varphi$ is generated by choosing a LTL template, $T$, the number of conjunctions, $N$, and then the proposition instantiations, $\boldsymbol{p}$ for each conjunct. The generative process for each of those components is as follows:

$$T \sim Categorical(w_T) \tag{5}$$
$$N \sim Geometric(\lambda) \tag{6}$$
$$p \sim Categorical(w_p) \tag{7}$$

$T$ is generated with respect to a categorical distribution with weights $w_T \in \mathbb{R}^k$ over the $k$ possible LTL templates. $w_T$ is a hyperparameter that the designer can set to assert preferences for mining certain types of templates over others (e.g., preferring templates with "global" operators than "until" operators).

The number of conjunctions, $N = |\{\boldsymbol{p}\}|$, is generated using a geometric distribution with a decay rate of $\lambda$. Thus, the the probability of $\varphi$ is reduced by $\lambda$ for each addition of a conjunct, incentivizing low-complexity specifications defined in terms of having a fewer number of conjunctions (which also implies fewer total propositions). This promotes conciseness and prevents over-fitting to the input traces (i.e., to avoid restating the input as a long, convoluted LTL formula).

Similar to the method used for template selection, we use a separate categorical distribution for selecting propositions $\boldsymbol{p}$ for each conjunct in $\varphi$. Propositions are generated with respect to the probability weights, $w_p \in \mathbb{R}^{|V|}$, defined for all $p$ in $\mathbf{V}$. The designer can likewise control $w_p$ to favor specifications instantiated with certain types of propositions over others. $w_p$ may be interpreted as the level of saliency of propositions for an application. (For example, propositions that are landmarks for planning problems (Hoffmann, Porteous, and Sebastia 2004), or a part of the causal links set (Veloso and Blythe 1994), may be deemed more important to express in plan explanations than other auxiliary state variables.) Several forms of variable importance, corresponding to the saliency of that importance in

an explanation, may be applied to set $w_p$. This opens the door to hypothesizing which propositions are most salient for a given domain, and generating explanations restricted to those propositions exclusively.

The full prior function, $P(\varphi)$, is evaluated as follows:

$$P(\varphi) = P(T)P(N)P(\{\boldsymbol{p}\}) \tag{8}$$

The derivation follows from the definition that $T$, $N$, $\{\boldsymbol{p}\}$ completely describe $\varphi$ (i.e. $P(\varphi \mid T, N, \{\boldsymbol{p}\}) = 1$), and the assumption that the three probability distributions are independent of each other. $P(T)$ and $P(N)$ are calculated using categorical and geometric distributions outlined in Equations 5 and 6, respectively. $P(\{\boldsymbol{p}\})$ denotes the probability of the full set of proposition instantiations (over all conjuncts); it is calculated by the average categorical weight, $w_p$, over all propositions. Formally:

$$P(\{\boldsymbol{p}\}) = \frac{\sum_{\boldsymbol{p} \in \{\boldsymbol{p}\}} \sum_{p \in \boldsymbol{p}} w_p}{N|\boldsymbol{p}|} \tag{9}$$

For example, with $\{\boldsymbol{p}\} = \{[a,b], [a,c], [b,c]\}$, and $w_a = 5/10$, $w_b = 4/10$, $w_c = 1/10$, $P(\{\boldsymbol{p}\}) = \frac{20/10}{6} = 1/3$.

**Likelihood Function**  The likelihood function $P(\mathbf{X} \mid \varphi)$ is the probability of observing the input sets of traces in the satisfying set $\pi_A$ and the non-satisfying set $\pi_B$ given the contrastive specification. The traces in $\pi_A$ and $\pi_B$ are generated by different solutions to the planning problem that satisfy the problem specification. As the problem specification is the only input needed to generate a set of plans, we assume that the individual traces are conditionally independent of each other, given the planning problem specification. With the conditional independence assumption, the likelihood can then be factored as follows:

$$P(\mathbf{X} \mid \varphi) = \prod_{i=1}^{|\boldsymbol{\pi_A}|} P(\pi_i|\varphi) \prod_{j=1}^{|\boldsymbol{\pi_B}|} P(\pi_j|\varphi) \tag{10}$$

LTL satisfaction checks are conducted over all traces belonging to sets $\boldsymbol{\pi_A}$ and $\boldsymbol{\pi_B}$; $P(\pi_i|\varphi)$ is set equal to $1 - \alpha$ if $\pi_i \models \varphi$, and $\alpha$ otherwise. Conversely, $P(\pi_j|\varphi)$ is set equal to $1 - \beta$ if $\pi_j \nvDash \varphi$, and $\beta$ otherwise. $\alpha$ and $\beta$ permit non-zero probability to traces not adhering to the constrastive explanation criterion, thereby providing robustness to noisy traces and outliers. $\alpha$ and $\beta$ may be set to different values to reflect the relative importance of the positive and negative sets (e.g., may be used to counteract imbalanced sets).

In order to perform LTL satisfaction checks on a trace, we follow the method developed by Lemieux et al. (2015), in which $\varphi$ is represented as a tree and each temporal operator is recursively evaluated according to its semantics. Since sub-trees of two different $\varphi$ may be identical, we memoize and re-use evaluation results to significantly speed up LTL satisfaction checks.

**Proposal Function**  Exact inference methods to find maximum a posterior (MAP) estimates, $\{\varphi^*\}$, are intractable. Thus we implement a Markov Chain Monte Carlo method, specifically the Metropolis-Hasting (MH) algorithm (Chib and Greenberg 1995), to iteratively draw samples whose

collection approximates the true posterior distribution. MH sampling requires a user-defined proposal function $F(\varphi'|\varphi)$ that samples a new candidate $\varphi'$ given the current $\varphi$. Our $F$ behaves similar to an $\epsilon$-greedy search, utilizing a drift kernel (i.e. a random walk) with a probability of 1-$\epsilon$ or sampling from the prior distribution (i.e. a restart) with a probability of $\epsilon$. The drift kernel operates by performing one of the following moves on the current candidate LTL $\varphi$:

- Remain within the current template $T$, *add* a new conjunct, and instantiate that conjunct with a randomly sampled $\boldsymbol{p}$ that is currently not in $\varphi$. The probability associated with this move, $Q_{add}$, is equal to $1/(|\mathbf{V}^{n_T}| - N)$.

- Remain within the current template $T$ and randomly *remove* one of the existing conjuncts. The probability associated with this move, $Q_{remove}$, is equal to $1/N$.

The selection between these two moves is conducted uniformly, though there is no issue with allowing the designer to weight one more likely than the other. Note that the drift kernel perturbs $\varphi$, but stays within the current template. $\varphi$ transitions to a new template (probabilistically) when choosing to sample from the prior.

The probability distribution associated with $F$, denoted by $Q(\varphi'|\varphi)$, is then outlined as follows:

$$
\begin{cases}
(1-\epsilon) \cdot 0.5 \cdot Q_{add}(\varphi'|\varphi) & \text{, drift (add move)} \\
(1-\epsilon) \cdot 0.5 \cdot Q_{remove}(\varphi'|\varphi) & \text{, drift (remove move)} \\
\epsilon \cdot P(\varphi') & \text{, sample prior function}
\end{cases}
$$

Our proposal function $F$ fulfills the ergodicity condition of the Markov process (the transition from any $\varphi$ to $\varphi'$ is aperiodic and occurs within a finite number of steps), thus asymptotically guarantees the sampling process from the true posterior distribution.

A new sample $\varphi'$ is accepted at every MH iteration with the following probability:

$$
\min\left(1, \frac{P(\varphi')P(\mathbf{X}|\varphi')Q(\varphi|\varphi')}{P(\varphi)P(\mathbf{X}|\varphi)Q(\varphi'|\varphi)}\right), \tag{11}
$$

The set of accepted samples approximates the true posterior, and the MAP estimates (the output $\{\varphi^*\}$) are determined from the relative frequencies of accepted samples.

## 5 Evaluations

### Derivation of Evaluation Dataset

We evaluated the effectiveness of our model for inferring contrastive explanations from sets of traces generated from a number of International Planning Competition (IPC) planning domains (Long and Fox 2003). The plan traces in $\boldsymbol{\pi_A}$ were generated by first injecting the ground truth $\varphi_{ground}$ into the original PDDL domain and problem files, enforcing valid plans on the modified domain/problem files to satisfy $\varphi_{ground}$. The LTL injection to create modified planning files was performed using the LTLFOND2FOND tool (Camacho et al. 2017). Second, a state-of-the-art top-k planner[1] (Katz

---

[1] An alternative would have been to use a *diverse* planner (Nguyen et al. 2012), but the existing ones do not support the required expressivity of conditional effects in the modified planning files.

---

et al. 2018) was used to produce a set of distinct, valid plans and their accompanying state execution traces.

Similarly, the above steps were repeated to generate execution traces for $\boldsymbol{\pi_B}$, wherein the negation of the ground truth specification, $\neg\varphi_{ground}$, was injected to the planning files, and then a set of traces was collected. Such a setup guarantees the existence of contrastive explanation solutions on $(\boldsymbol{\pi_A}, \boldsymbol{\pi_B})$, which includes (but is not limited to) $\varphi_{ground}$. We collected twenty traces for each set.

We evaluated our model using six different IPC benchmark domains, containing problems related to mission planning, vehicle routing, and resource allocation. For each of these domains, we tested three different problem instances of increasing vocabulary size, and on twenty randomly generated $\varphi_{ground}$ specifications for each problem instance.

### Experiment Details

For each test case, $\varphi_{ground}$ was randomly generated using one of the seven LTL templates listed in Table 1; thus the hypothesis space $\boldsymbol{\Phi}$ was set to include all possible specifications over the predefined templates. The categorical distribution weights, $w_T$ and $w_p$, were set to be uniform. Other hyperparameters were set as follows: $\alpha = \beta = 0.01$, to put equal importance of positive and negative sets, $\lambda = 0.7$ to penalize $\varphi$ for every additional conjunct, and $\epsilon = 0.2$ to apply $\epsilon$-greedy search in the the proposal function. We ran the MH sampler with $num_{MH} = 2,000$ iterations with the first 300 used as a burn-in period. These hyperparameters were set apriori, similar to how a wide range of probablistic graphical models are designed. However, our experimental results were found to be robust to the various settings of these parameters.

### Model Comparisons

We evaluated our model against the SAT-based miner developed by Neider and Gavran (2018), the state-of-the-art for extracting contrastive LTL specifications. We also evaluated our model against brute force enumeration, a common approach employed by existing LTL miners used for summarization (Yang et al. 2006; Lemieux, Park, and Beschastnikh 2015). Because enumerating through full space of $\boldsymbol{\Phi}$ would result in a time out, we tested *delimited* enumeration with only a random subset of brute force samples. This baseline selects a random subset of size $num_{brute}$ from $\boldsymbol{\Phi}$. Then, a function proportional to the posterior distribution (numerator in Equation 4) is evaluated for each of the samples to determine $\{\varphi^*\}$. $num_{brute}$ was set equal to $num_{MH}$ to enable a fair baseline in terms of having the same amount of allotted computation.

## 6 Results and Discussion

Table 2 shows the inference results on the tested domains and on problem instances of varying complexity (reflected by an increase in $|\boldsymbol{V}|$). For evaluation, we measured $M = |\{\varphi^*\}|$, the number of unique contrastive explanations extracted by the different approaches, along with the explanations' accuracy. Each domain-problem combination row shows the average statistics over twenty $\varphi_{ground}$ test cases.

| Domain | $|V|$ | Our Model | | Enumeration | | Neider & Gavran | |
|---|---|---|---|---|---|---|---|
| | | *M* | *Acc* | *M* | *Acc* | *M* | *TimeOut* |
| blocks-world | 16 | **3.1** | 0.97 | 0.9 | 0.96 | 1 | **10** / 20 |
| | 25 | **8.1** | 1.00 | 0.8 | 0.95 | 1 | **10** / 20 |
| | 55 | **30.2** | 0.99 | 0.7 | 0.83 | 0 | **20** / 20 |
| storage | 11 | **4.4** | 0.94 | 1.7 | 0.97 | 1 | **16** / 20 |
| | 20 | **11.5** | 0.97 | 1.0 | 0.94 | 1 | **14** / 20 |
| | 42 | **31.5** | 0.98 | 0.9 | 0.93 | 1 | **11** / 20 |
| satellite | 17 | **22.3** | 1.00 | 3.7 | 0.97 | 1 | **6** / 20 |
| | 37 | **28.0** | 0.98 | 0.9 | 0.85 | 1 | **8** / 20 |
| | 50 | **84.6** | 0.97 | 1.0 | 0.94 | 1 | **8** / 20 |
| zeno-travel | 18 | **3.7** | 0.99 | 2.1 | 0.99 | 1 | **8** / 20 |
| | 22 | **21.0** | 1.00 | 1.7 | 0.99 | 1 | **8** / 20 |
| | 40 | **78.6** | 0.99 | 1.5 | 0.99 | 1 | **3** / 20 |
| TPP | 14 | **7.5** | 1.00 | 1.1 | 0.95 | 0 | **20** / 20 |
| | 18 | **9.7** | 1.00 | 0.8 | 0.95 | 0 | **20** / 20 |
| | 36 | **114.5** | 0.99 | 2.3 | 0.96 | 1 | **14** / 20 |
| rovers | 20 | **24.5** | 1.00 | 2.3 | 0.99 | 1 | **5** / 20 |
| | 22 | **28.3** | 1.00 | 2.3 | 0.99 | 1 | **10** / 20 |
| | 28 | **18.6** | 0.98 | 1.0 | 0.97 | 1 | **11** / 20 |

Table 2: Inference results on extracting contrastive explanations across different approaches. Each row reports the averages across twenty $\varphi_{ground}$ test cases. *M* denotes the number of unique contrastive explanations, and *Acc* reports their average accuracy.

High *M* and high accuracy across all domain-problem combinations demonstrate how our probabilistic model was able to generate multiple, near-perfect contrastive explanations. The solution set $\{\varphi^*\}$ almost always included $\varphi_{ground}$. Our model outperformed the baseline and the state-of-the-art miner by producing more contrastive explanations within an allotted amount of computation / runtime.

The runtime for our model and the delimited enumeration baseline with 2,000 samples ranged between 1.2–4.7 seconds (increase in $|V|$ only had marginal effect on the runtime). The SAT-based miner by Neider and Gavran often failed to generate a solution within a five minute cutoff (see the number of its timeout cases in the last column of Table 2). The prior work can only output a single $\varphi^*$, which frequently took on a form of $\mathbf{F}p_i$. It did not scale well to problems that required more complex $\varphi$ as solutions. This is because increasing the "depth" of $\varphi$ (the number of temporal / Boolean operators and propositions) exponentially increased the size of the compiled SAT problem. In our experiments, the prior work timed out for problems requiring solutions with depth $\geq 3$ (note that $\mathbf{F}p_i$ has depth of 2).

Figure 2 compares the search efficiency between our model and the delimited enumeration baseline. By employing more informed search with the MH sampler, our Bayesian inference model discovered contrastive explanations with high accuracy with much fewer iterations and lower variance than compared to the baseline. The trend was consistent across all test domains.

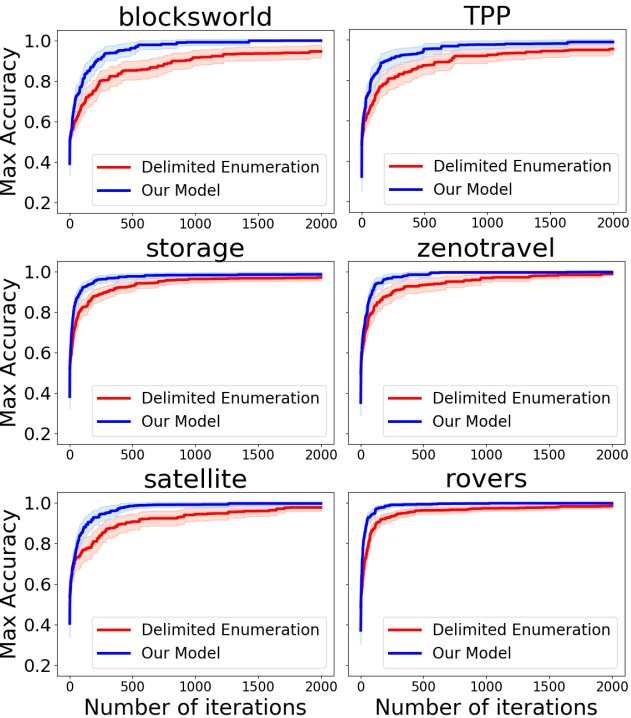

Figure 2: The max accuracy of $\{\varphi^*\}$ with respect to the number of sampling iterations. The comparison is shown for both the MH sampler and the delimited enumeration baseline. Each domain subplot shows the averages all three problem instances and all twenty $\varphi_{ground}$ test cases. 95% confidence intervals are displayed.

**Robustness to Noisy Input** In order to test robustness, we perturbed the input $\mathbf{X}$ by randomly swapping traces between $\pi_A$ and $\pi_B$. For example, a noise rate of 0.2 would swap 20% of the traces, where the accuracy of $\varphi_{ground}$ on the perturbed data, $\widetilde{\mathbf{X}} = (\widetilde{\pi_A}, \widetilde{\pi_B})$, would evaluate to 0.8 (note that it may be possible to discover other $\varphi$ that achieve better accuracy on $\widetilde{\mathbf{X}}$). The MAP estimates inferred from $\widetilde{\mathbf{X}}$, $\{\widetilde{\varphi^*}\}$, were evaluated on the original input $\mathbf{X}$ to assess any loss of ability to provide contrast.

Figure 3 shows the average accuracy of $\{\widetilde{\varphi^*}\}$, evaluated on both $\widetilde{\mathbf{X}}$ and $\mathbf{X}$, across varying noise rate. Even at a moderate noise rate of 0.25, the inferred $\widetilde{\varphi^*}$s were able to maintain an average accuracy greater than 0.9 on $\mathbf{X}$. Such a threshold is promising for real-world applications. The robustness did start to sharply decline as noise rate increased past 0.4. For all test cases, the Neider and Gavran miner failed to generate a solution for anything with a noise rate $\geq 0.1$.

**Solution Space of Contrastive Explanations** Large values of $M$ signify how there are often various ways to express how plan traces differ using the LTL semantics. Some LTL specifications are logically dependent. For example, the global template subsumes both the stability and the eventuality template. LTL specifications may also be related through substitutions of propositions. For example, on

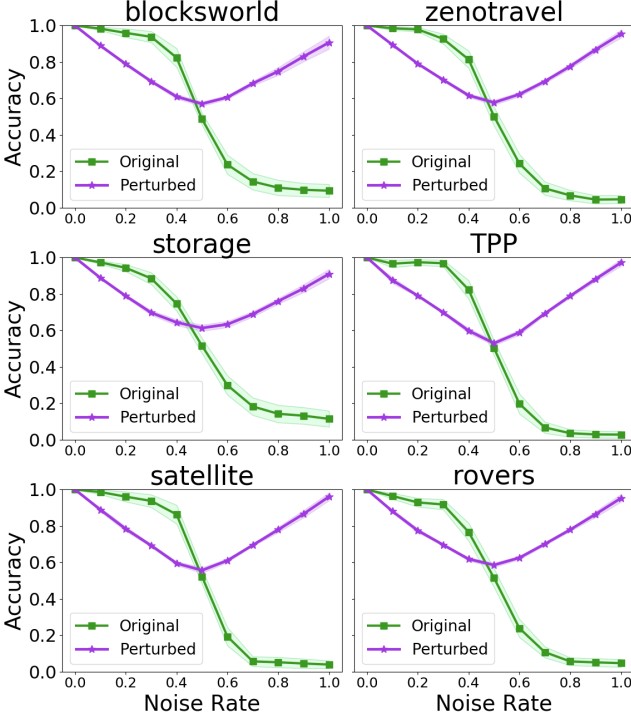

Figure 3: The accuracy of $\widetilde{\varphi^*}$ with respect to increasing noise rate. $\widetilde{\varphi^*}$ is inferred from the perturbed, noisy data and then is evaluated (generalized) on the original input $\mathbf{X}$. Each domain subplot shows the averages across all three problem instances and all twenty $\varphi_{ground}$ test cases. 95% confidence intervals are displayed.

problems where holding a block is a prerequisite to placing it onto a table, $\varphi_1 = \mathbf{F}(holding\_A) \wedge \mathbf{F}(holding\_B)$ will be satisfied in concert with the satisfaction of $\varphi_2 = \mathbf{F}(ontable\_A) \wedge \mathbf{F}(ontable\_B)$. For contrastive explanation, however, one needs to be mindful of both positive and negative sides of satisfaction which affect the accuracy. Relations like template subsumptions or precondition / effect pairs should not be simply favored during search without understanding that the converse may not hold and may result in worse accuracy.

For a contrastive $\varphi$, it is possible to create a new contrastive $\varphi'$ that includes stationary propositions or tautologies specific to the planning problem. For example, if $\varphi_1 = \mathbf{F}(holding\_A) \wedge \mathbf{F}(holding\_B)$ is a contrastive explanation, so is $\varphi_3 = \mathbf{F}(holding\_A) \wedge \mathbf{F}(holding\_B) \wedge \mathbf{F}(earth\_is\_round)$. Our posterior distribution assigns a lower probability to $\varphi_3$ than $\varphi_1$ based on the decay rate on the number of conjunctions. Also, tautologies by themselves cannot be contrastive explanations, because they can never be dissatisfied. The output of our model appropriately excluded such vacuous explanations.

Table 2 shows how $M$ generally increased as $|\mathbf{V}|$ increased. This opens up interesting research avenues for determining a minimal set of $\{\varphi\}$. Assessing logical dependence or metric space between two arbitrary LTL specifications, however, is non-trivial.

**Evaluation on Real-world Inspired Domain** We applied our inference model on a large force exercise (LFE) domain, which simulate air-combat games used to train pilots. Through the use of Joint Semi-Automated Forces environment (Anastasiou 2006), realistic aircraft behavior and their state execution traces were collected for the mission objective of "gain and maintain air superiority." A total of 24 instances (i.e. traces) of LFEs were separated into positive and negative sets by a subject matter expert. The detail of the input was as follows: $|\boldsymbol{\pi}_A|$=16, $|\boldsymbol{\pi}_B|$=8, $|\mathbf{V}|$=15, and the average length of traces involved 11 time steps.

Within a second (2,000 samples), our model generated ten unique contrastive explanations, all with accuracy of 0.96. $\varphi_1^* = \mathbf{G}(attrition < 0.25) \wedge \mathbf{G}(striker\ not\ shot)$ represents how friendly attrition rate should be always less than 25% and that the striker aircraft should never be shot upon. $\varphi_2^* = (attrition < 0.25)\ \mathbf{U}\ (weapon\ release)$ asserts how friendly attrition rate has to be less than 25% before releasing the weapon. The model also inferred rules of the environment, for example, asserting that propositions $(attrition < 0.75)$ and $(attrition < 0.50)$ precede $(attrition < 0.25)$ (which makes sense because attrition can only increase throughout the mission). After discussion with the expert, we discovered that the model could not generate the perfect contrastive $\varphi_{ground}$, because it required having multiple conjuncts that incorporate different LTL templates (which is not part of our defined solution space). Nevertheless, the generated explanations were consistent with the expert's interpretation of achieving the mission objective of air superiority.

## 7 Conclusion

We have presented a probabilistic Bayesian model to infer contrastive LTL specifications describing how two sets of plan traces differ. Our model generates multiple contrastive explanations more efficiently than the state-of-the-art and demonstrates robustness to noisy input. It also provides a principled approach to incorporate various forms of prior knowledge or preferences during search. It can serve as a strong foundation that can be naturally extended to multiple input sets by repeating the algorithm for all pairwise or one-vs.-rest comparisons.

Interesting avenues for future work include gauging the saliency of propositions, as well as deriving a minimal set of contrastive explanations. Furthermore, we seek to test the model in human-in-the-loop settings, with the goal of understanding the relationship between different planning heuristics for the saliency of propositions (e.g. landmarks and causal links) to their actual explicability when the explanation is communicated to a human.

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
