# OpenReview forum: "Bayesian Inference of Temporal Specifications to Explain How Plans Differ"
_icaps-conference.org/ICAPS/2019/Workshop/XAIP — XAIP 2019_

### Official Review · AnonReviewer1 · 2019-04-25
**Inferring LTL specifications of differences between sets of plan traces**

**Rating:** 3
**Confidence:** 2

**Review:**

This is a mature piece of technical work addressing the problem of finding LTL characterizations of the differences between two given sets of plan traces. The paper introduces and evaluates an algorithmic improvement over previous efforts in this direction. The techniques and results are novel and significant at a scale more than suited to a workshop.

What remains largely unclear though is the connection to XAI. My understanding of the term/the area is that we try to explain the decisions taken by an AI system. The presented problem, instead, is concerned with summarizing/explaining the differences between two sets of possible decisions. I can certainly agree that this may be somehow related. But how? The authors do discuss this point, and attempt to justify their work as XAI, but I don't follow their points/claims in this regard.

The authors say:

"Our work on using LTL for contrastive explanations directly contributes to exploring how we can answer the following roadmap questions for XAIP (Fox, Long, and Magazzeni 2017): “why did you do that? why didn’t you do something else (that I would have done)?"

It remains completely unclear to me inhowfar this is supposed to be the case. How does a summary of the difference between two plans explain why the planner decided to propose one or the other plan? How does it explain why not something else was decided? The best connection I can see is that such a summary may serve as a tool, making it easier for a human user to try to answer/make sense of these questions herself.

A little further down, similarly:

"According to Elzein (2018), a contrastive explanation describes “why event A occurred as opposed to some alternative event B.” In our problem, events A and B represent two sets of plan traces (can be seen as traces generated from
different systems or different group behavior). The form of why may be expressed in various ways (Lombrozo 2006); our choice is to define it according to the plans’ satisfaction of a constraint."

It remains completely unclear to me inhowfar a constraint (satisfied by one plan set but not the other) is a form of "why" in this context. It's much more a form of "what", isn't it?

After these two recommendable but all-too-vague discussions of XAIP, the paper basically proceeds from the formal definition, specifying algorithms and evaluating these. This is nice technical work, but the connection to XAIP is never meaningfully taken up again so far as I can see. At the least, the authors could try to get back to this discussion based on some of the example domains they use in their evaluation? Exemplifying, in that manner, in which way they mean to address "why" questions in contrastive explanation?

Overall, a nice piece of technical work with a somewhat tenuous connection to XAIP. Can be accepted. If it is, I'd appreciate if the authors could address the points I raise above, in the paper and its presentation at the workshop.

---

### Official Review · AnonReviewer2 · 2019-05-08
**Review 2**

**Rating:** 4
**Confidence:** 2

**Review:**

The paper describes an general approach for generating differences between positive and negative plan sets using LTL.  These resulting differences can then be used to generate contrastive explanations.  After detailing the approach, which can be intractible, the paper describes a mechanism to use Beyesian inferrence with search to identify plan differences.  In a set of experiments on 6 benchmark problems, they paper evaluates the approach and demonstrates that it is able to find more differences than an existing approach.  Finally, the paper summarizes the use of this approach for a real-world problem.

Generally, the paper is clear and well motivated.  I think it would make a fine contribution to the workshop.  It is especially noteworthy that this work presents a general method for producing contrastive explanations, which would benefit the XAIP community considerably.  The authors may want to add the work of Miller 2018 (previous XAI workshop) and 2019 (AIJ 267:1-38) regarding contrastive explanations.  Also, the paper should probably cite the recently accepted ICAPS 2019 paper by Chakraborti et al. called "Explicability? Legibility? Predictability? Transparency? Privacy? Security? The emerging landscape of interpretable behavior"

Minor comments:
 - Consider a different variable from X for LTL or Bayesian Inference.  At first I was confused by how the same X could be used for both.
 - Section 5 should have a segue rather than go straight into a subsection.

---

### Decision · Program_Chairs · 2019-05-15

**Decision:**

Accept

**Comment:**

The reviewers agree to accept. Please address all review criticism as best possible for the final paper version and its presentation at the workshop. Looking forward to discuss your work at the workshop!